# Reconstruction of Large Soft Tissue Defects in the Distal Lower Extremity: Free Chain-Linked Bilateral Anterolateral Thigh Perforator Flaps versus Extended Latissimus Dorsi Musculocutaneous Flaps

**DOI:** 10.3390/jpm12091400

**Published:** 2022-08-29

**Authors:** Jiqiang He, Gunel Guliyeva, Panfeng Wu, Liming Qing, Fang Yu, Juyu Tang

**Affiliations:** 1Department of Hand & Microsurgery, Xiangya Hospital of Central South University, 87 Xiangya Road, Changsha 410008, China; 2Department of Plastic Surgery, Mayo Clinic, Jacksonville, FL 32224, USA

**Keywords:** chain-linked bilateral anterolateral thigh perforator flap (alt), extended latissimus dorsi musculocutaneous flap (ld), large soft-tissue defects, lower extremity, reconstructive surgical procedures/methods, surgical flaps

## Abstract

Background: Reconstruction of the large soft-tissue defects in the lower extremity still constitutes a challenge for plastic surgeons. This retrospective study was conducted to compare the surgical and clinical outcomes of the chain-linked bilateral anterolateral thigh perforator flaps and extended latissimus dorsi musculocutaneous flap in the reconstruction of the large soft tissue defects of the lower extremity. Methods: From January 2012 to December 2021, 34 patients aged between 20 and 66 years received chain-linked bilateral anterolateral thigh perforator flaps (15 cases) or extended latissimus dorsi musculocutaneous flaps (19 cases) for the reconstruction of extensive soft-tissue defects in the lower extremity. The two groups were homogeneous in terms of age, etiology, comorbidities, and flap area. In addition, the intraoperative data, outcomes, complications, and long-term follow-up results were collected and analyzed. Results: The extended latissimus dorsi musculocutaneous flap group had a shorter operation time (271.8 ± 59.5 min vs. 429.6 ± 51.9 min), harvest time (58.9 ± 24.8 min vs. 152.7 ± 41.4 min), and anastomosis time (27.2 ± 10.4 min vs. 53.7 ± 8.1 min) than the chain-linked bilateral anterolateral thigh perforator flaps group (*p* < 0.05). Based on patient self-assessment, the donor site temporary muscle weakness in the extended latissimus dorsi musculocutaneous flap group was significantly more than that in the chain-linked bilateral anterolateral thigh perforator flaps group (*p* < 0.05). Conclusion: Both methods can repair large defects and restore the function of the injured limbs at a single stage. However, considering the operation time and flap-harvesting time, the authors recommend the extended latissimus dorsi musculocutaneous flap, especially for those who cannot tolerate a prolonged surgery.

## 1. Introduction

Reconstruction of extensive soft-tissue defects in the lower extremity is a challenge for plastic surgery [1,2]. This kind of wound usually requires flap transplantation to cover critical structures such as exposed joints, bones, and tendons. Single-stage reconstruction of large soft-tissue defects is preferred as it allows the restoration of the function of the lower extremities [3,4]. However, finding a suitable donor site to harvest a flap of adequate length is difficult. For this purpose, the use of combined or sequential chimeric flap transplantation has been reported to be reliable [5,6,7].

Nevertheless, these flaps require a high level of experience and are technically challenging. Both the chain-linked bilateral anterolateral thigh perforator (ALT) flap and the latissimus dorsi musculocutaneous (LD) flap are used to repair large wounds [8,9]. Nonetheless, in the literature, there is no evidence regarding the superiority of any of these flaps in the reconstruction of extensive defects in the lower extremity. Thus, the purpose of our study was to compare the outcomes of the chain-linked bilateral ALT flaps and the extended LD flap in the reconstruction of large soft tissue defects in the distal lower extremity.

## 2. Materials and Methods

### 2.1. Patients

Between January 2012 and December 2021, 34 patients aged from 20 to 66 years received chain-linked bilateral ALT flaps or extended LD flaps for the reconstruction of extensive soft-tissue defects. While fifteen defects were repaired by chain-linked bilateral ALT flaps, nineteen patients were reconstructed by extended LD flap. Demographic factors, intraoperative data, outcomes, complications, and long-term follow-up results were collected and analyzed. Written informed consent was obtained from the patients. The study was approved by the Ethics Committee of Xiangya Hospital.

Inclusion criteria included (1) patients who had extensive soft tissue defects in the distal lower extremity; (2) the defect area was over 300 cm^2^, which could not be repaired by a skin graft or local flap; (3) patients who agreed to receive either chain-linked bilateral ALT flaps or extended LD flap for one-stage reconstruction.

Exclusion criteria: (1) lost to follow-up patients; (2) previous failed flap surgery; (3) patients with severe comorbidities

All the operations were completed by a single surgeon and his team using the following surgical technique.

### 2.2. Surgical Method

Initial thorough debridement was performed in an outside institution before patients were sent to our hospital for subsequent reconstruction. All 30 trauma patients had negative pressure wound therapy.

Preoperatively, the arterial system of the affected lower extremity was evaluated, most often by ultrasonography or angiography. Our purpose was to exclude the presence of the anatomical variations of major vessels or underlying vascular disease existed. Additionally, this technique also helped us check the continuity of the main arterial trunk of the injured lower extremity.

In the chain-linked bilateral ALT group (Figure 1), the perforators were detected and marked with a handheld doppler or CTA. The defect template was split into two parts to design bilateral ALT flaps. The harvest method was the same as reported in our previous study [10]. Briefly, the chain-linked bilateral ALT flaps were harvested using the retrograde dissection technique. The skin flap elevation plane was above the deep fascia, proceeding medially from the lateral side. Several additional perforators were preserved until the main perforator was identified. Then, the dissection was performed meticulously to isolate the vascular pedicle. The perforator was traced back to the main trunk of the descending branch of the lateral circumflex femoral vessels (LCFV). After the bilateral ALT flaps were dissected entirely, these flaps were placed side-by-side to repair the large soft-tissue defects. The proximal end of the LCFV was anastomosed to the recipient’s vessels, and the distal end of the descending branch was anastomosed to another ALT flap. After the flap was transferred to the defect, the donor site was closed directly with complete hemostasis and reliable drainage.

In the extended LD group (Figure 2), the design based on the defect was placed on the lateral chest and upper back. The following surgical procedures were used to harvest the extended LD flap [11].

First, the patient was placed in a lateral position. Then, based on the design, we incised the lateral skin and subcutaneous tissue, then exposed the space between the latissimus dorsi muscle and serratus anterior. Afterward, we incised the medial skin and subcutaneous tissue and detached the subcutaneous tissue of the latissimus dorsi muscle. Instead of harvesting the whole latissimus dorsi muscle, we harvested the appropriate size of latissimus dorsi muscle based on the extent of the wounds. After the flap was harvested and transferred to the defect, the thoracodorsal vessels were anastomosed to the recipient’s vessels. The residual exposed latissimus dorsi muscle and the wound could then be covered with split-thickness skin grafts. Finally, the donor site was closed primarily with a cosmetic suture.

### 2.3. Evaluation of Outcomes

We evaluated the flap characteristics and intra-operative data, flap complications and outcomes, and recorded any complications. Functional results were assessed by questionnaires using the Lower Extremity Functional Scale (LEFS): based on a scale from 0 (poor) to 100 (excellent) [12]. The long-term cosmetic results of the recipient sites were evaluated subjectively by the patients and objectively by a blinded third-party observer. We evaluated the cosmetic appearance of the recipient sites using the Visual Analog Scale (VAS) score from 0 (unsatisfactory) to 10 (excellent). In our study, 0–2 points were considered unsatisfactory, 3–5 were fair, 6–8 were satisfactory, and 9–10 were excellent. Donor site scar hyperplasia was assessed by the Vancouver Scar Scale (VSS) [13]. We also evaluated the donor site functional damage.

### 2.4. Statistical Analysis

Quantitative data are expressed as means ± standards (standard deviation) and compared using the Student’s t-test. Qualitative data are expressed as numbers or percentages and compared using the χ^2^ test and Fisher’s exact test. Statistical analysis was performed by SPSS 23.0 software (SPSS, Inc., Chicago, IL, USA). *p* < 0.05 was considered statistically significant.

## 3. Results

To compare the outcomes of chain-linked bilateral ALT flaps and extended LD flap reconstruction, 34 patients with large skin and soft-tissue defects were included in this retrospective study (Table 1). A total of 14 males and 1 female received chain-linked bilateral ALT flaps, and 15 males and 4 females received an extended LD flap. The average age for the chain-linked bilateral ALT flaps group was 40.9 ± 12.0 years, while the average age for the extended LD flap group was 42.5 ± 13.3 years. There were no significant differences in age, sex, risk factors, wound sizes, or etiology between the two groups (*p* > 0.05).

### 3.1. Flap Characteristics and Intra-Operative Data

The flap characteristics and intra-operative data are summarized in Table 2. The mean flap size was 440.3 ± 96.2 cm^2^ in the chain-linked bilateral ALT flaps group and 391.8 ± 113.8 cm^2^ in the extended LD flap group (*p* > 0.05). The extended LD flap group had a shorter flap elevation time (58.9 ± 24.8 min vs. 152.7 ± 41.4 min), vascular anastomosis time (27.2 ± 10.4 min vs. 53.7 ± 8.1 min), and operation time (271.8 ± 59.5 min vs. 429.6 ± 51.9 min) than the chain-linked bilateral ALT flaps group (*p* < 0.05). There were no significant differences between the recipient vessels of the two groups (*p* > 0.05). 

### 3.2. Flap Complications and Outcomes

The flap complications and outcomes were analyzed (Table 3). All except one flap from the chain-linked bilateral ALT flaps group survived. Total necrosis occurred in one of the chain-linked bilateral ALT flaps group, and none occurred in the extended LD flap group. Partial necrosis was observed in three cases in the chain-linked bilateral ALT flaps group and two cases in the extended LD flap group. There was no difference in the factors causing flap necrosis between the two groups (arterial insufficiency, venous congestion, hematoma, and infection (*p* > 0.05)).

For one case with total flap necrosis, we used the extended LD flap to salvage the limb. Dressing changes were done if partial flap necrosis is located at the flap margins. Otherwise, skin grafting or local flap transfer were performed.

### 3.3. Long-Term Follow-Up Results

The long-term follow-up results are listed in Table 4. The chain-linked bilateral ALT flaps group showed four donor site scar hyperplasias, but the extended LD flap group had ten. Nevertheless, the difference was not statistically significant (*p* > 0.05).

In the overall cosmetic evaluations, the subjective scoring showed that three cases (20.0%) of the chain-linked bilateral ALT flaps group patients rated their appearance as excellent, four cases (26.7%) as satisfactory, six cases (40.0%) as fair, and two cases (13.3%) as unsatisfactory. In the extended LD flap group, two cases (10.5%) of patients rated their appearance as excellent, three cases (15.8%) as satisfactory, eight cases (42.1%) as fair, and six cases (31.6%) as unsatisfactory. The results showed no significant differences in satisfaction of appearance between the chain-linked bilateral ALT flaps group and the extended LD flap group (*p* > 0.05). This was also observed for the ratings of the blinded third-party observers (*p* > 0.05). Besides, there was no significant difference between the chain-linked bilateral ALT flaps group and the extended LD flap group for the functional recovery of the lower extremity (*p* > 0.05).

### 3.4. Subjective Assessment of Donor Site Function

Patient self-evaluations of donor-site function are listed in Table 5. Only three patients in the chain-linked bilateral ALT flaps group reported a temporary muscular weakness, while 13 patients in the extended LD had temporary muscular weakness. The results showed that the temporary muscular weakness of the donor site of the patients in the extended LD flap group was significantly higher than that in the chain-linked bilateral ALT flaps group (*p* < 0.05).

None reported permanent muscular weakness or limitation of knee joint movement in the chain-linked bilateral ALT flaps group. In the extended LD flap group, three cases of permanent muscular weakness and two cases of limitations in the shoulder’s range of motion were noted, but the differences were not statistically significant between these two groups (*p* > 0.05).

## 4. Discussion

Large soft-tissue defects of the lower extremity are common in road traffic accidents. Such injuries usually involve extensive bone and tendon exposure, which is difficult to handle properly and can lead to lower extremity amputation [14]. One-stage flap repair has the advantage of preventing infections and restoring lower extremity function [15]. However, the severity of injury generally precludes the use of local flaps. Free flaps, especially chain-linked bilateral ALT flaps and extended LD flaps, are feasible options for repairing large skin and soft tissue defects of the lower extremity. However, there is no “gold standard.”

The LD flap is easily elevated and has a large surface area, which is a reliable and versatile method for free tissue transfer. It was first described as a pedicled musculocutaneous flap by Tansini in 1906 [16]. After Baudet described the use of the latissimus dorsi as a free tissue transfer, it became one of the most popular flaps for free tissue transfer [17]. With the development of the thoracodorsal artery perforator flap, the LD flap is being gradually disregarded due to its donor site morbidities [18]. However, for those with large skin and soft-tissue defects, the LD flap still can be a useful option to salvage the extremity [19]. Ozkan et al. [20] used the LD flap to reconstruct large defects of the lower extremity. Their technique demonstrates that LD flap transplantation is highly successful in limb salvage procedures both from a functional and cosmetic standpoint.

Although the LD flap can be used for more extensive wounds, the donor site scar is also a problem [21]. Sometimes harvesting a large flap requires skin grafting to close the donor site. To avoid this, we modified the design and tried to harvest a narrow part of the skin flap with an additional muscle flap, which enabled us to close the donor site directly. With this modified design, the incidence of donor site scar hyperplasia was not significantly increased when compared with chain-linked bilateral ALT flaps. Additionally, split-skin grafted muscle flaps have been claimed to be more stable than fasciocutaneous flaps because they are easily contoured to the recipient area [22,23,24]. In our study, we did not find any significant difference between chain-linked bilateral ALT flaps and modified LD flaps in the cosmetic evaluation (subjective and objective).

On the other hand, donor site functional damage is the main disadvantage of the LD flap. Russell et al. [25] found that elderly patients and those in the early postoperative period were more likely to report disability. In the long follow-up, various synergistic muscles can undertake more of the latissimus dorsi muscle’s function with exercise. In another systematic review [26], the authors found that patients had little difficulty in daily activities, but significant difficulties in sports and art activities. Nevertheless, in the case of a large defect, we believe that the benefit of the lower extremity salvage outweighs this drawback.

The anterolateral thigh flap was first described by Song et al. [27], and it has a large cutaneous area, long vascular pedicle, acceptable donor-site morbidity, and is a workhouse flap in the reconstructive field. Recently, our team has published their experience with different versions of ALT flaps such as flow-through, double-paddle, sequential chimeric, and chain-linked bilateral ALT for different kinds of wounds [28,29,30,31]. For large defects, chain-linked bilateral ALT is a reliable option. For example, Qing et al. [32] reported using combined chain-linked ALT flaps for hand and forearm defects with large skin and soft-tissue defects. In their study, the mean size of the total flaps was 419.6 cm^2^. Nonetheless, this technique is challenging, time-consuming, and exhaustive. Our study found that the outcome of chain-linked bilateral ALT flaps was also good with less donor site morbidity, but the operation time, flap harvest time, and vascular anastomosis time were significantly longer than extended LD flap.

Although the above two flaps have known advantages and disadvantages, a study comparing the outcomes of these two methods in repairing large skin and soft tissue defects has not been published. Our retrospective study found no significant difference in the surgical outcome between the two methods, but the operation time and the flap harvest time in the extended LD flap was shorter. When comparing the donor site complications, we found that the extended LD flap had more donor site problems in the early stage, but the patient’s daily life was not significantly affected during the long-time follow-up. A previous study has compared the outcome between the ALT flap and LD flap in lower extremity reconstruction, and they found that the ALT flap should be preferred due to its minimal donor site mobility [33]. Nevertheless, this study only considered the single flap surgery for small or medium defects, and for large defects, we think the extended LD flap should be selected.

This study had the following limitations: The number of cases is limited. Besides the short follow-up of the donor site, we still lack the objective and quantitative method to compare the sacrifice of the donor site and the benefit of the recipient site. In addition, this was a single-center retrospective study, and other reconstructive methods such as thoracic–umbilical flaps and sequential chimeric flaps were not included due to fewer cases being performed in our department. We are aiming to compare more options for larger skin and soft-tissue defects and conduct a prospective multi-center study in the future.

## 5. Conclusions

In summary, both methods can, in one stage, repair large defects and restore the function of the injured limbs. However, considering the operation time and flap harvest time, an extended LD flap is recommended, especially for patients who cannot endure a prolonged surgery.

## Figures and Tables

**Figure 1 jpm-12-01400-f001:**
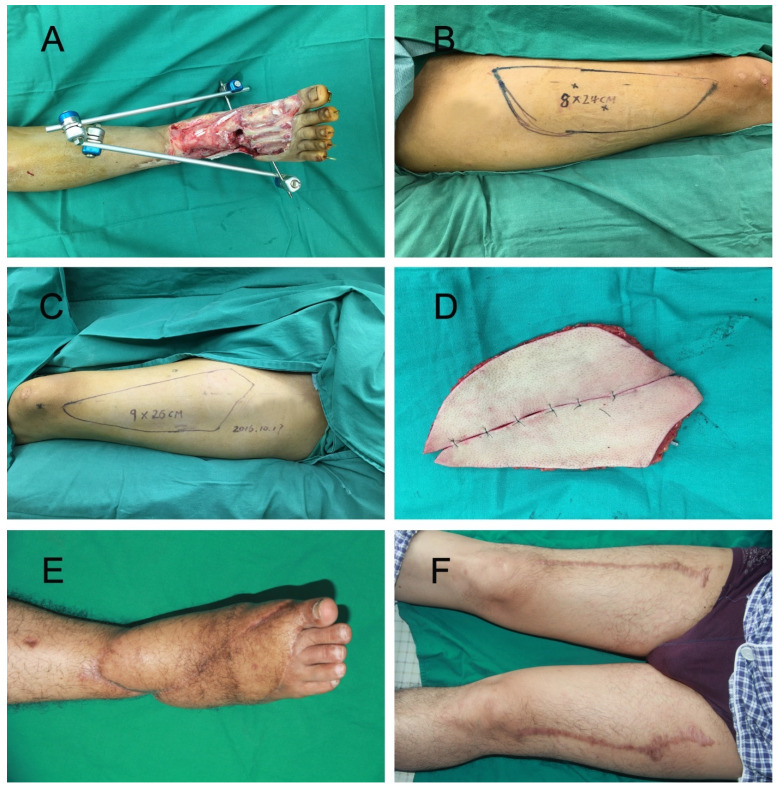
(**A**) A 30-year old male sustained injuries to the right ankle and foot in a traffic accident. Radical debridement left a large wound on the right lower extremity. (**B**) Right ALT flap with 24 × 8 cm^2^ was designed. (**C**) Left ALT flap with 26 × 9 cm^2^ was designed. (**D**) Chain-linked bilateral ALT flaps were harvested. (**E**,**F**) Recipient and donor site at the 12-month follow-up.

**Figure 2 jpm-12-01400-f002:**
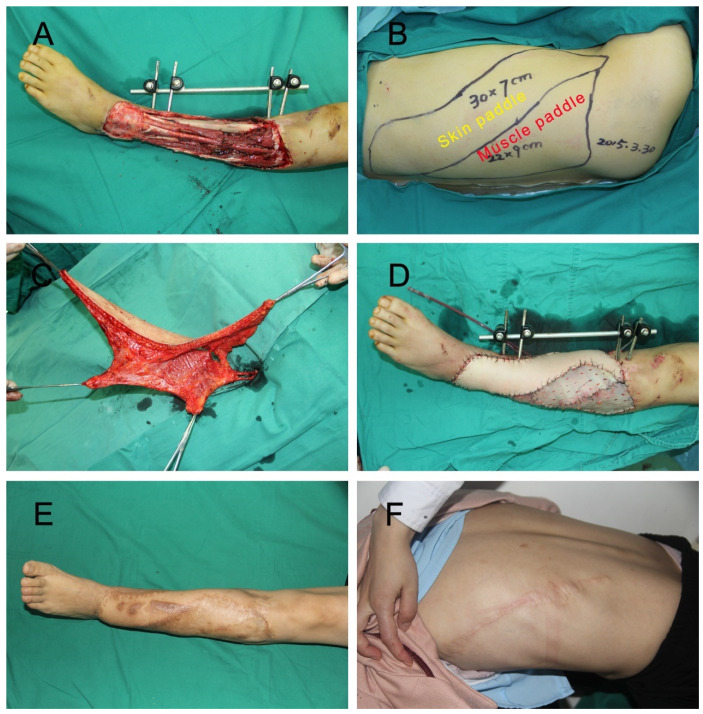
(**A**) A 28-year old female injured in a traffic accident that caused a large soft-tissue defect on the left lower leg. (**B**) Extended LD flap design. (**C**) Harvest of the flap. (**D**) The extended LD flap was used to cover the large defects. The exposed part of the latissimus dorsi muscle flap and remaining lower extremity wounds repaired with split-skin graft. (**E**,**F**) Recipient and donor site at the 19-month follow-up.

**Table 1 jpm-12-01400-t001:** Demographic data.

Variable	Chain-Linked Bilateral ALT Group (N = 15)	Extended LD Group (N = 19)	*p* Value ^†^
Age (years) ^#^			
Age (years)	40.9 ± 12.0	42.5 ± 13.3	0.728
Sex			0.355
Male	14	15	
Female	1	4	
Age ≥ 60	1	2	1.000
Smokers	7	6	0.484
Type 2 diabetes	2	8	0.128
Atherosclerosis detected ^&^	3	6	0.697
Wound size (cm^2^) ^#^	406.4 ± 91.9	359.4 ± 108.9	0.191
Etiology			1.000
Traumatic	14	16	
Others *	1	3	
Follow-up, mo ^#^	18.8 ± 11.2	14.9 ± 6.0	0.207

ALT, anterolateral thigh perforator flap. LD, latissimus dorsi musculocutaneous flap. ^#^ *t*-test. ^&^ Diagnosed either on computed tomographic angiography or arteriography. * Keloid or hypertrophic scar, chronic ulcer. ^†^ Two-sided Fisher’s exact test.

**Table 2 jpm-12-01400-t002:** Flap characteristics and intraoperative data.

Variable	Chain-Linked Bilateral ALT Group (N = 15)	Extended LD Group (N = 19)	*p* Value ^#^
Flap size (cm^2^)	440.3 ± 96.2	391.8 ± 113.8	0.196
Flap elevation time, min	152.7 ± 41.4	58.9 ± 24.8	*p* < 0.001
Vascular anastomosis time, min	53.7 ± 8.1	27.2 ± 10.4	*p* < 0.001
Operation time, min	429.6 ± 51.9	271.8 ± 59.5	*p* < 0.001
Recipients vessels ^†^			0.728
Anterior tibial system	10	11	
Posterior tibial system	5	8	

ALT, anterolateral thigh perforator flap. LD, latissimus dorsi musculocutaneous flap. ^#^ *t*-test. ^†^ Two-sided Fisher’s exact test.

**Table 3 jpm-12-01400-t003:** Flap complications and outcomes.

Variable	Chain-Linked Bilateral ALT Group (N = 15)	Extended LD Group (N = 19)	*p* Value ^†^
Flap success rate	93.3%	100%	0.441
Flap-related complications			0.370
Total flap necrosis	1	0	
Partial flap necrosis	3	2	
Factors of causing flap necrosis			
Artery insufficiency	2	0	0.187
Venous congestion	0	1	1.000
Infection	1	1	1.000
Hematomas	1	0	0.441

ALT, anterolateral thigh perforator flap. LD, latissimus dorsi musculocutaneous flap. ^†^ Two-sided Fisher’s exact test.

**Table 4 jpm-12-01400-t004:** Long-term follow up results.

Variable	Chain-Linked Bilateral ALT Group (N = 15)	Extended LD Group (N = 19)	*p* Value ^†^
Donor site scar hyperplasia ^#^	4	10	0.171
Cosmetic evaluation			
Subjectively ^a^			0.288 ^c^
Excellent	3	2	
Satisfactory	4	3	
Fair	6	8	
Unsatisfactory	2	6	
Objectively ^b^			0.165 ^c^
Excellent	4	2	
Satisfactory	5	4	
Fair	5	8	
Unsatisfactory	1	5	
Lower extremity functional evaluation			0.613 ^c^
Excellent	8	7	
Satisfactory	6	9	
Fair	1	3	
Poor	0	0	

ALT, anterolateral thigh perforator flap. LD, latissimus dorsi musculocutaneous flap. ^#^ Diagnosed by the Vancouver Scar Scale. ^†^ Two-sided Fisher’s exact test. ^a^ Patients themselves. ^b^ Blinded third-party. ^c^ Excellent and satisfactory rate.

**Table 5 jpm-12-01400-t005:** Subjective assessment of donor site function.

Variable	Chain-Linked Bilateral ALT Group (N = 15)	Extended LD Group (N = 19)	*p* Value ^†^
Temporary muscular weakness	3	13	0.007
Permanent muscular weakness	0	3	0.238
Limitations of joint movement	0	2	0.492

ALT, anterolateral thigh perforator flap. LD, latissimus dorsi musculocutaneous flap. ^†^ Two-sided Fisher’s exact test.

## Data Availability

Data sharing is not applicable to this article.

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
