# Peer review of "Reconstruction of Large Soft Tissue Defects in the Distal Lower Extremity: Free Chain-Linked Bilateral Anterolateral Thigh Perforator Flaps versus Extended Latissimus Dorsi Musculocutaneous Flaps"

_jpm, 2022, doi:10.3390/jpm12091400_

Round 1
Reviewer 1 Report
Dear authors,
The subject of the study is very interesting.
I suggest you to improve the study method by establishing more clearly the indications and the criteria of choosing the surgical techniques used.
I suggest you, to show, eventually, the particularities of the cases according to which the use of one flap or other was chosen.
The informations for authors regarding the writing of bibliographic indices in the manuscript must be reviewed
Author Response
RC 1. Dear authors, the subject of the study is very interesting. I suggest you to improve the study method by establishing more clearly the indications and the criteria for choosing the surgical techniques used.
Re: Thank you for your kind comment! We have elaborated on the inclusion criteria to make the study clear to readers.
RC 2. I suggest you, to show, eventually, the particularities of the cases according to which the use of one flap or other was chosen.
Re: Thank you for your suggestion! Unfortunately, this is a limitation of our study. As this study has a retrospective design, we could not identify the reasons for the flap choice.
RC3.The information for authors regarding the writing of bibliographic indices in the manuscript must be reviewed
Re: Thank you for pointing this out! Reference has been revised.
Reviewer 2 Report
Dear authors,
Your manuscript has not the proper template for JPM.
In addition your study is too similar with this one:
He J, Qing L, Wu P et al. Large wounds reconstruction of the lower extremity with combined latissimus dorsi musculocutaneous flap and flow-through anterolateral thigh perforator flap transfer. Microsurgery 2021; 41: 533-542.
Author Response
RC 1. Dear authors, your manuscript has not the proper template for JPM.
Re: Thank you for your comment! We have modified the manuscript according to the JPM guidelines.
RC 2. In addition your study is too similar with this one: He J, Qing L, Wu P et al. Large wounds reconstruction of the lower extremity with combined latissimus dorsi musculocutaneous flap and flow-through anterolateral thigh perforator flap transfer. Microsurgery 2021; 41: 533-542.
Re: Thank you for your comment! These papers indeed share some similarities. However, the previous paper is a case series to introduce our experience with large wound reconstruction with the combined transfer of latissimus dorsi musculocutaneous and flow-through ALT flap.
On the other hand, the current study is a comparative study of free chain-linked bilateral anterolateral thigh perforator flaps and extended latissimus dorsi musculocutaneous flaps. The aim of this study is to understand if either flap is better suited for large soft tissue defects. Moreover, the surgical procedure described in these two papers differ.
Reviewer 3 Report
no suggestions
Author Response
No special comments.